# Sialic Acids on Tumor Cells Modulate IgA Therapy by Neutrophils via Inhibitory Receptors Siglec-7 and Siglec-9

**DOI:** 10.3390/cancers15133405

**Published:** 2023-06-29

**Authors:** Chilam Chan, Marta Lustig, J. H. Marco Jansen, Laura Garcia Villagrasa, Leon Raymakers, Lois A. Daamen, Thomas Valerius, Geert van Tetering, Jeanette H. W. Leusen

**Affiliations:** 1Center for Translational Immunology, University Medical Center Utrecht, Heidelberglaan 100, 3584 CX Utrecht, The Netherlands; c.l.chan@umcutrecht.nl (C.C.); m.jansen-8@umcutrecht.nl (J.H.M.J.); l.garciavillagrasa@students.uu.nl (L.G.V.); l.raijmakers@umcutrecht.nl (L.R.); geert@gtbiotechconsultancy.com (G.v.T.); 2Division of Stem Cell Transplantation and Immunotherapy, Department of Medicine II, Christian Albrechts University Kiel and University Medical Center Schleswig-Holstein, Campus Kiel, 24105 Kiel, Germany; marta.lustig@uksh.de (M.L.); t.valerius@med2.uni-kiel.de (T.V.); 3Imaging Division, University Medical Center Utrecht, Utrecht University, 3584 CX Utrecht, The Netherlands; l.a.daamen-3@umcutrecht.nl; 4Department of Surgery, Regional Academic Cancer Center Utrecht, UMC Utrecht Cancer Center, St. Antonius Hospital Nieuwegein, Utrecht University, 3584 CX Utrecht, The Netherlands

**Keywords:** IgA, myeloid checkpoint inhibition, neutrophils, tumor sialylation, CD47/SIRPα

## Abstract

**Simple Summary:**

In cancer patients, immunotherapy using targeted antibodies is often not effective in the long term due to resistance mechanisms developed by cancer cells. Tumors often overexpress certain molecules on the cell surface, that bind to specific receptors on immune cells, such as myeloid cells that can inhibit the immune response against cancer cells. One of such interactions involves sialic acid molecules on cancer cells and specific receptors called Siglecs on myeloid cells. This interaction prevents neutrophils (a type of myeloid cell) from effectively killing tumor cells. To improve the effectiveness of immunotherapy using a specific type of antibody called IgA, we investigated ways to disrupt the interaction between sialic acid and Siglecs. Removal of sialic acids from the tumor cells, a process called desialylation, enhanced the ability of neutrophils to kill tumor cells in the presence of IgA antibodies. Additionally, combining desialylation with blocking the CD47/SIRPα interaction, that was also reported as an inhibitory interaction on neutrophils, further improved the immune response. The study suggests that a combination of blocking different inhibitory interactions, such as CD47/SIRPα and sialic acids/Siglec, may be necessary to optimize cancer immunotherapy, considering the ways in which tumor cells evade the immune system.

**Abstract:**

Immunotherapy with targeted therapeutic antibodies is often ineffective in long-term responses in cancer patients due to resistance mechanisms such as overexpression of checkpoint molecules. Similar to T lymphocytes, myeloid immune cells express inhibitory checkpoint receptors that interact with ligands overexpressed on cancer cells, contributing to treatment resistance. While CD47/SIRPα-axis inhibitors in combination with IgA therapy have shown promise, complete tumor eradication remains a challenge, indicating the presence of other checkpoints. We investigated hypersialylation on the tumor cell surface as a potential myeloid checkpoint and found that hypersialylated cancer cells inhibit neutrophil-mediated tumor killing through interactions with sialic acid-binding immunoglobulin-like lectins (Siglecs). To enhance antibody-dependent cellular cytotoxicity (ADCC) using IgA as therapeutic, we explored strategies to disrupt the interaction between tumor cell sialoglycans and Siglecs expressed on neutrophils. We identified Siglec-9 as the primary inhibitory receptor, with Siglec-7 also playing a role to a lesser extent. Blocking Siglec-9 enhanced IgA-mediated ADCC by neutrophils. Concurrent expression of multiple checkpoint ligands necessitated a multi-checkpoint-blocking approach. In certain cancer cell lines, combining CD47 blockade with desialylation improved IgA-mediated ADCC, effectively overcoming resistance that remained when blocking only one checkpoint interaction. Our findings suggest that a combination of CD47 blockade and desialylation may be necessary to optimize cancer immunotherapy, considering the upregulation of checkpoint molecules by tumor cells to evade immune surveillance.

## 1. Introduction

In recent years, IgA antibodies have emerged as a potential isotype for tumor eradication. Although IgA is primarily known for its critical role as secretory IgA in mucous membrane immune functions, monomeric IgA antibodies have been successfully adapted to target tumor associated antigens such as HER2, EGFR, CD20, and GD2 in various in vitro and in vivo studies [1,2,3,4]. IgA immune complexes engage efficiently with the Fc receptor for IgA, FcαRI (CD89), found on various immune cells such as neutrophils, eosinophils, monocytes, and macrophages [5]. FcαRI crosslinking induces signaling through its associated FcR γ-chains, bearing immunoreceptor tyrosine-based activation motifs (ITAMs) in its intracellular domain. Phosphorylated ITAM then acts as a docking site for Scr and Syk family protein tyrosine kinases to induce a signaling cascade, which results in immune-mediated effector functions such as antibody-dependent cellular phagocytosis (ADCP) and ADCC [4,6]. The latter is essential in the anti-tumor response of neutrophils mediated by IgA, a mechanism known as trogoptosis [7].

Cancer cells have developed a variety of strategies to avoid immune surveillance, including the overexpression of checkpoint molecules [8]. These are typically immunoinhibitory receptor ligands that inhibit immune responses via conserved immunoreceptor tyrosine-based inhibitory motifs (ITIMs). As a result, they establish an immunosuppressive tumor microenvironment (TME), which encourages immune evasion and tumor growth [9]. The first checkpoints to be identified were T cell checkpoints including cytotoxic T lymphocyte-associated protein 4 (CTLA-4) and programmed cell death protein 1 (PD-1)/programmed death-ligand 1 (PD-L1). Expression of immune checkpoints is not limited to T cells. In recent years, novel immunoinhibitory receptors on myeloid cells have been discovered, giving rise to a new class of checkpoints known as myeloid checkpoints. One of the checkpoints that affects neutrophil trogoptosis and ADCC is the CD47/signal regulatory protein alpha (SIRPα) axis. Overexpression of CD47 in solid and hematological cancer cells is associated with a poor prognosis [10,11]. Studies have shown reduced cytotoxicity upon CD47 binding to SIRPα on neutrophils, which could be reversed by interrupting the interaction [12,13,14]. Treffers et al. (2020) found that knocking out CD47 could inhibit tumor growth in a long-term A431 xenograft mouse model [14]. However, total tumor eradication was not achieved, indicating the presence of alternative mechanisms driving therapeutic resistance.

One of these mechanisms could be found in aberrant sialylation of cancer cells, which function as a negative regulator of immune cells. Malignant cells overexpress sialic acid-containing carbohydrates (sialoglycans) with terminal sialic acids as a result of deregulation of the glycan synthesis pathway, leading to tumor cell hypersialylation [15,16,17]. Sialic acids are recognized by Siglecs expressed on immune cells. In neutrophils, Siglec-3, -5, -7, -9, and -14 are expressed. However, Siglec-14 lacks any tyrosine-based motif, whereas Siglec-3, -5, -7, and -9 are ITIM-bearing receptors that, when activated by their sialic acid ligands, suppress the immune response in a similar manner to CD47/SIRPα engagement [8,18]. Despite the fact that all Siglecs bind to terminal sialic acids, recognition by specific Siglec receptors is dependent on the sialoglycan structure beyond the terminal sialic acid. Furthermore, recognition is heavily dependent on the linkage orientation to the core glycan and subterminal carbohydrate moieties. Recent studies have identified specific sialoglycan moieties found on malignant cells to be ligands for Siglec-7 and -9 [19,20,21,22,23]. This, however, needs to be explored further. Although studies on NK cells, T cells, and monocytes/macrophages have demonstrated that Siglecs can contribute to tumor growth, only a limited number of studies have addressed the role of Siglecs in neutrophils [20,23,24,25,26]. A recent study conducted by Lustig et al. provided evidence supporting the suppressive role of Siglec interaction in neutrophils in the context of cancer therapy [23]. While the group investigated the impact of sialic acid/Siglec-9 inhibition on IgG-induced neutrophil cytotoxicity, the potential effects on IgA-induced neutrophil cytotoxicity remain to be explored.

We studied the effect of checkpoint inhibition on IgA therapy beyond the CD47/SIRPα axis focusing on neutrophils. While we show that the CD47/SIRPα axis is a key regulator in tumor growth, tumors frequently exploit multiple mechanisms to avoid immune surveillance. Here, we confirm that a wide range of tumor cells are hypersialylated. Desialylation of tumor cells improved IgA-mediated neutrophil killing in in vitro systems and in an in vivo mouse model. Our findings show that, although multiple Siglecs are expressed on neutrophils, the majority of the human cancer cell lines tested only express Siglec-7 and Siglec-9 ligands. Using a Siglec-9 blocking antibody, we confirmed that Siglec-9 is the primary Siglec regulating IgA-mediated neutrophil ADCC in hypersialylated tumor cells. In addition, we observed a minor involvement of Siglec-7 in regulating this process in one of the cell lines tested. Furthermore, we were able to further increase the anti-tumor response by blocking both the CD47/SIRPα axis and the sialic acid/Siglec axis. Our findings support the notion that Siglecs in IgA therapy for cancer could be the next novel myeloid checkpoint target.

## 2. Material and Methods

### 2.1. Cell Culture and Cell Lines

Unless otherwise stated, all tumor cell lines were purchased from ATCC and cultured at 37 °C in a humidified incubator containing 5% CO_2_. A431-HER2, AsPC-1, Ba/F3, BxPC-3, DLD-1 (gift from Thomas Valerius group, University of Kiel), MCF-7, NCI-H1838, Panc 10.05, and SK-BR-3 were cultured in RPMI 1640 (Thermo Fisher. Gibco, Grand Island, NY, USA) supplemented with 10% fetal calf serum (FCS) and 100 U/mL penicillin–streptomycin (Pen/Strep, Gibco, life technologies). Ba/F3 cells were additionally supplemented with 0.2 ng/mL recombinant mouse IL-3 (Immunotools), and A431-HER2 cells were cultured under 0.5 μg/mL puromycin (Sigma, Marlborough, MA, USA) selection. MDA-MB-175, MDA-MB-231, and MDA-MB-468 were cultured in Leibovitz’s L-15 (Thermo Fisher. Gibco) supplemented with 10% FCS and 100 U/mL Pen/Strep without CO_2_ exchange. Capan-2 cells were maintained in McCoy’s 5A (Thermo Fisher. Gibco) supplemented with 10% FCS and 100 U/mL Pen/Strep. CFPAC-1 cells were maintained in IMDM (Thermo Fisher. Gibco) supplemented with 10% FCS and 100 U/mL Pen/Strep. A431-HER2-GNE knockout (KO) and scrambled (Scr) cells were cultured in OptiMEM I reduced-serum medium (Thermo Fisher. Gibco) supplemented with 100 U/mL Pen/Strep and 0.5 μg/mL of puromycin. A431-Scr and A431-CD47 KO were described previously [14]. Cells were not cultured past 20 passages, and they were regularly tested for mycoplasma contamination using a Mycoalert mycoplasma detection kit (Lonza, Basel, Switzerland).

### 2.2. ^51^Cr Release ADCC Assay

Sialic acids on target cells were hydrolyzed via pre-treatment with 0.1 U/mL of neuraminidase (NEU) in serum-free RPMI medium for 1 h in a shaking incubator at 125 rpm and 37 °C. Neuraminidase activity was inactivated by adding complete RPMI medium. ADCC assays were carried out as described previously [27]. In brief, target cells were labeled with 100 μCi chromium-51 (PerkinElmer) per million cells for at least 2 h at 37 °C and 5% CO_2_. Next, cells were washed thrice with medium. Subsequently, to interrupt the CD47/SIRPα axis, target cells were pre-treated with 10 μg/mL IgG1 PGLALA SIRPα fusion protein, hereafter called SIRPα fusion protein, for 30 min at room temperature. Peripheral blood was obtained from healthy donors at the UMC Utrecht, and leftover patient blood was obtained from treatment-naïve patients with primary pancreatic cancer at the Regional Academic Cancer Center Utrecht, as part of the PRIMOPANC project (institutional project number V0000304). Human polymorphonuclear leukocytes (PMN) were isolated from peripheral blood by first performing a Ficoll (GE Healthcare, Chicago, IL, USA) density gradient centrifugation to remove all peripheral blood mononuclear cells (PBMC), leaving only red blood cells (RBC) and granulocytes in the pellet. Subsequently, red blood cells were lysed using RBC Lysis Buffer (Biolegend, San Diego, CA, USA), and only the remaining PMNs were used in the ADCC assays.

PMNs were added to the target cells in a 40:1 effector-to-target (E:T) ratio with antibodies at the concentrations specified in the experiment. After 4 h of incubation at 37 °C in a humidified incubator containing 5% CO_2_, the plate was centrifuged, and the supernatant was transferred to a LumaPlate (PerkinElmer, Waltham, MA, USA) to be measured on a beta-gamma counter for radioactive scintillation (in cpm) (PerkinElmer). Specific lysis was calculated using the formula: ((Experimental cpm − basal cpm)/(maximal cpm − basal cpm)) × 100. The maximum cpm was determined by treating target cells with 5% Triton X-100 (Sigma-Aldrich), and the baseline cpm was determined by chromium release from target cells in the absence of antibodies and effector cells.

### 2.3. Antibodies and Reagents

The α2,3-linked sialyl residues on cells were stained with 5 μg/mL biotinylated *Maackia Amurensis* Lectin II (MAL II, Vector laboratories) and detected with APC-conjugated Streptavidin (eBioscience). Neuraminidase *Clostridium Perfingens* (Sigma Aldrich), *Vibrio Cholerae* (Merck) *Arthrobacter ureafaciens* (Sigma Aldrich), and *Salmonella Typhimurium* (SIALST) were used at 0.1 U/mL to hydrolyze terminal sialic acid. Surface EGFR expression was determined with an anti-EGFR antibody (clone AY13, Biolegend). In the CD47 binding competition assay CD47-PE (CC2C6, Biolegend) was used to detect displacement of the SIRPα fusion protein. Detection of Siglec ligands on tumor cells was performed using 10 μg/mL recombinant human Siglec-5, -7, and -9 Fc chimera protein (Bio-Techne). IgA3.0 antibodies, hereafter called IgA, against EGFR (Cetuximab), EpCAM (HeING), and HER2 (Trastuzumab) were produced and purified in house as described in Stip et al. (2023, in press). CD47-blocking, SIRPα fusion protein was produced and purified in house as described in Chernyavska et al. (2022) [28]. IgG2 FcKO Siglec-9 blocking antibody (mAbA) was produced and purified in house [23]. 

### 2.4. Flow Cytometry

Flow cytometric analysis was used to determine antibody binding to assess protein expression and antibody binding. A total of 100,000 cells were stained with antibody in FACS buffer (PBS, 0.01% BSA, 0.01% Na-Azide) for 45 min. on ice. To hydrolyze sialic acids, cells were pre-treated with 0.1 U/mL neuraminidase for 1 h in a shaking incubator at 125 rpm, 37 °C, and 8% CO_2_. To determine the antigen density per cell, we performed a QIFIKIT (Agilent/Dako) analysis. The assay was performed according to the manufacturer’s instructions. Cells were stained with a saturating concentration of 10 μg/mL unconjugated mouse IgG monoclonal antibody directed against EGFR (AY-13, Biolegend), HER2 (H2Mab-77, Biolegend), EpCAM (9C4, Biolegend), or CD47 (CC2C6, Biolegend). Measurements were performed on BD FACS Canto II.

### 2.5. CRISPR/Cas9 of GNE

The biosynthesis of sialic acid was disrupted in A431-HER2 cells by knocking out a key enzyme in this pathway. Knockout of UDP-N-acetylglucosamine 2-epimerase/N-acetylmannosamine kinase (UDP-GlcNAc 2-epimerase/ManNAc kinase; GNE) was performed with CRISPR/Cas9 using a single guide RNA (sgRNA, AGGATTTACACGGCCACCTG, IDT) targeting *GNE* or a scrambled negative control (IDT). A431-HER2 cells were prepared for electroporation with the Neon Transfection System (Thermo Fisher) according to the manufacturer’s instructions. Cells were pulsed a single time with 1400 V and 30 ms, KO efficiency was determined by flow cytometry, and the low MAL II lectin cell population was sorted to enrich the A431-HER2-GNE KO population.

### 2.6. Short i.p. In Vivo Model

Mice were housed in the University of Utrecht’s Central Laboratory Animal Research Facility. Female human FcαRI (CD89) transgenic (Tg) mice were used in the experiment, which were generated at the UMC Utrecht and backcrossed on a SCID (NOD.CB17-Prkdcscid/scid/Rj) background. Transgene-negative (non-Tg) littermates were used for the solvent control (PBS) group. All mice were housed and bred at Janvier Labs in Paris, France, and ranged in age from 11 to 34 weeks. Mice were acclimatized for at least one week prior to the start of the experiment after being transported to Utrecht. Mice were housed in groups in a temperature-controlled room with a 12:12 h light:dark cycle, with food and water available ad libitum. Mice were randomized based on age prior to the start of the experiment, and the treatment and analysis were performed blindly.

Three days prior to injection, MDA-MB-468 cells were treated with 100 μM Sialyltransferase Inhibitor (3Fax-Peracetyl Neu5Ac, Sigma Aldrich) or DMSO. Following this, the cells were labeled for 15 min. at room temperature in the dark with either 8 μM CellTrace Violet (Thermo Fisher) or 1 μM CellTrace CSFE fluorescent dye (Thermo Fisher), respectively. Untreated EGFR-negative murine Ba/F3 cells were labeled similarly with 0.4 μM CellTrace Violet. Next, 200 μl of PBS containing a total of 9 million cells was injected into the peritoneum after the cells were mixed in a 1:1:1 ratio. A single intraperitoneal injection of 2.5 μg IgA3.0 Cetuximab or PBS (100 μL) was administered immediately after tumor cell inoculation. After 16 h, all mice (*n* = 6) were sacrificed, and the tumor cells were obtained by performing a peritoneal lavage with PBS containing 5 mmol/L EDTA. The absolute number of tumor cells were determined using Trucount absolute counting tubes (BD Biosciences). The IgA killing capacity was determined by calculating the ratio of MDA-MB-468 and control Ba/F3 cells in the peritoneum determined by flow cytometry. Moreover, sialic acid expression on the tumor cells was determined with 5 μg/mL biotinylated MAL II (Vector laboratories) and detected with APC-conjugated Streptavidin (eBioscience, San Diego, CA, USA). Neutrophils in the peritoneum were determined using anti-CD45 (30-F11, Biolegend) and anti-Ly-6G (1A8, Biolegend) antibodies, and their relative count was normalized to a constant number of beads (Invitrogen, Waltham, MA, USA).

### 2.7. Data Processing and Statistical Analyses

Flow cytometry analysis was performed in FlowJo (TreeStar, Woodburn, OR, USA). Statistical analyses were performed using GraphPad Prism 9.3.0 (GraphPad Software Inc., San Diego, CA, USA) and represented as mean ± standard deviation (SD) or standard error of mean (SEM) where a *p*-value < 0.05 was considered a significant difference.

## 3. Results

To study the overcoming of resistance to therapeutic IgA therapy in cancer cells through CD47 blockade, we investigated the combination of IgA antibodies and disruption of the CD47/SIRPα axis, a well-studied myeloid checkpoint pathway, in the context of tumor cell killing by neutrophils. To disrupt the inhibitory axis, we generated a CD47 KO of the epidermoid carcinoma cell line, A431. Flow cytometry analysis confirmed the abolished CD47 expression in A431-CD47 KO while showing no significant effect on the EGFR expression level (Figure 1A). We next assessed the killing capacity of IgA anti-EGFR (Cetuximab) by neutrophils targeting A431 cells using a ^51^Cr release ADCC assay. Consistent with previous findings, killing of A431-CD47 KO cells was significantly enhanced compared to that of A431-Scr cells (Figure 1B) [14]. 

To validate the inhibitory role of CD47 expression in neutrophil-mediated killing of tumor cells, we screened a large number of tumor cell lines and evaluated the expression levels of common tumors associated antigen targets, including CD47 (Figure 1C), as well as the expression of EGFR, EpCAM, and HER2 (Figure 1D). Previously, Brandsma et al. (2015) [27] suggested that IgA-mediated lysis is most effective with medium to high levels of target expression. Therefore, we proceeded to focus on cell lines with a target expression level greater than 2 × 10^5^ molecules per cell.

CD47 expression level was generally consistent across cell lines except for CFPAC-1, which exhibited a higher CD47 expression (Figure 1C). However, we found no clear correlation between CD47 expression level and resistance to IgA therapy. As such, pancreatic cell lines Capan-2 (CD47 low), Panc 10.05 (CD47 medium), and CFPAC-1 (CD47 high) express similar levels of EpCAM and varying levels of CD47. Notably, when we blocked CD47 on CFPAC-1 using a high-affinity SIRPα fusion protein, the cells remained resistant to IgA-mediated lysis (Figure 1C–E). In fact, we observed a specific lysis of only 10% compared to at least three times higher lysis of the other pancreatic cells. To verify that the SIRPα fusion protein was capable of blocking such high CD47 expression, we performed a competitive assay, which showed complete blocking of the CD47 binding epitope at a concentration of 10 μg/mL, the same concentration used in our ADCC assay (Appendix A). These findings suggest that other myeloid checkpoints, in addition to CD47, may be present on CFPAC-1 cells and contribute to its resistance to neutrophil-mediated lysis.

### 3.1. Hypersialylation of Tumor Cells Inhibit Killing by Neutrophils

Using MAL II lectin detection by flow cytometry, we confirmed expression of sialoglycans on all cancer cell lines in our panel (Figure 2A) and observed varying degrees of sialylation. Interestingly, CFPAC-1, which was previously shown to be very resistant to IgA-mediated ADCC, is highly sialylated, suggesting a possible explanation for the resistance to killing by neutrophils. To confirm the immunosuppressive role of hypersialylation, we treated the SK-BR-3 breast cancer cell line, which has medium sialoglycan expression, with 10 mM N-acetylneuraminic acid (human sialic acid, Neu5Ac) for 48 h to increase surface sialylation. Our analysis confirmed an upregulation of α2,3-linked sialic acids in the treated group (Figure 2B), and importantly, we observed a significant reduction in IgA-mediated ADCC by neutrophils (Figure 2C). Subsequently, we assessed if removing sialic acids by neuraminidases would lead to improved ADCC. Neuraminidases are a family of exoglycosidases that hydrolyze terminal sialic acids from sialoglycans. We tested neuraminidases from different bacterial strains for this study: neuraminidase from *Clostridium perfringens* (NEU-CP), *Vibrio cholerae* (NEU-VC), *Arthrobacter ureafaciens* (NEU-AU), and *Salmonella typhimurium* (Neu-ST), all of which cleave terminal sialic-acid residues that are α2,3-, α2,6-, or α2,8-linked (as stated by the manufacturer’s datasheet). To activate the enzyme, tumor cells were incubated with neuraminidase at 0.1 U/mL for 1 h maintained at 37 °C. Flow cytometry analysis using MAL II lectin demonstrated a consistent and comparable reduction in α2,3-linked sialic acids on SK-BR-3 cells, regardless of the bacterial source (Figure 2D and Appendix A). Furthermore, we observed effective desialylation of cell lines expressing higher numbers of sialic acids, including A431 and CFPAC-1 (Figure 2D). Following this, we aimed to investigate the effect of neuraminidase treatment on IgA-mediated ADCC by neutrophils. To this end, we evaluated various cell lines, including SK-BR-3, A431-HER2, and CFPAC-1. We found that neuraminidase treatment significantly improved tumor cell lysis in both SK-BR-3 and A431-HER2 cells, as marked by the enhanced ADCC capacity (Figure 2E,F). No differences were observed between the different bacterial sources of neuraminidase, indicating that all were effective in desialylation (Appendix A). Furthermore, we attempted to use neuraminidase treatment to improve IgA killing of the previously mentioned resistant cell line CFPAC-1. However, we found that neuraminidase treatment alone was not sufficient to improve IgA-mediated killing (Figure 2G). Given the concurrent high expression of CD47, we tried to combine CD47 blockade with neuraminidase treatment. Interestingly, we observed significantly increased IgA-mediated killing, with a nearly three-fold increase in lysis compared to IgA with CD47 blockade alone. Importantly, checkpoint inhibitors alone did not induce killing of the tumor cells, and significant lysis was only observed in the presence of IgA (Figure 2H). Moreover, to assess the feasibility of using checkpoint inhibition strategies for cancer patients, we compared PMNs from a patient with cholangiocarcinoma or pancreatic ductal adenocarcinoma (PDAC) to those of healthy donors. Our results showed no significant differences in Siglec-7, Siglec-9, and SIRPα expression (Figure 2I). Moreover, PMNs obtained from cancer patients retain their ability to effectively kill tumor cells and are responsive to checkpoint inhibition treatments (Figure 2J). Notably, when compared to healthy controls, PMNs derived from the cancer patient had comparable, if not superior, ADCC capacity. Taken together, neuraminidase treatment is an effective method for enhancing IgA-mediated ADCC in tumor cells with high sialic acid content, such as SK-BR-3, A431, and potentially other similar cell lines. However, the combination of neuraminidase treatment and CD47 blockade may be required for enhancing IgA-mediated killing of resistant tumor cells with high CD47 expression. Ultimately, checkpoint inhibition therapies may represent a promising treatment option for cancer patients.

### 3.2. Knockout GNE Impairs Sialic Acid Biosynthesis

To confirm the effect of desialylation, we genetically modified A431 cells that had previously been transduced to overexpress HER2. We used CRISPR/Cas9 to knock out *GNE*. The GNE gene encodes for the bifunctional enzyme UDP-GlcNAc 2-epimerase/ManNAc kinase [29]. Both are required for the biosynthesis of Neu5Ac. A431-HER2-GNE KO reduced MAL II binding compared to the scramble (Scr) control, while EGFR expression was unaffected (Figure 3A). Desialylation could be partly reversed by incorporating Neu5Ac into the medium (Figure 3B). In an ADCC assay, lysis of A431-HER2-GNE KO cells was comparable to that of A431-HER2-Scr cells treated with neuraminidase (Figure 3C). To verify our hypothesis, we added Neu5Ac to A431-GNE KO-HER2 cells, which reduced ADCC significantly, consistent with the increase in sialic acids on the cell surface. These findings support the immunosuppressive properties of sialic acids on tumor cells and show that genetically deleting GNE can overcome this.

### 3.3. Siglec-7 and Siglec-9 as Key Receptors on Neutrophils Interacting with Hypersialylated Tumors

We investigated the specific Siglecs expressed on neutrophils that may contribute to the observed immune suppression. Various Siglecs were examined, and flow cytometry analysis confirmed high expression of Siglec-5 and Siglec-9, while Siglec-3 and Siglec-7 showed low expression (Figure 4A). Ligand screening with recombinant Siglec-Fc proteins identified the presence of ligands for Siglec-7 and Siglec-9 on our cancer cell lines (Figure 4B). Siglec-7 appeared to predominantly mediate interactions with CFPAC-1 cells, given the absence of Siglec-9 ligands. In contrast, the other cell lines, with higher levels of Siglec-9 ligands, likely engage Siglec-9. Interestingly, the low expression of Siglec-7 on PMNs could explain the observed improvement in IgA-mediated ADCC only in combination with CD47 blockade. To confirm the involvement of sialic acids in these interactions, we treated CFPAC-1 and SK-BR-3 cells with neuraminidase, resulting in reduced binding of Siglec-7-Fc and Siglec-9-Fc, respectively. (Figure 4C). Moreover, blocking Siglec-9 on SK-BR-3 cells in an ADCC assay significantly increased lysis, reaching approximately 80% of the maximum lysis observed with neuraminidase treatment (Figure 4D).

### 3.4. Enhancing IgA-Mediated ADCC in Resistant Tumor Cells through a Combination of Siglec-9 and CD47 Blockade

We investigated the effect of Siglec-9 blockade on three different cell lines, SK-BR-3, A431-HER2, and MDA-MB-468. In previous experiments, we showed that targeting CD47 or desialylation separately improved killing by IgA with neutrophils. Here, we combined blocking both the CD47/SIRPα axis and Siglec-9/Sialic acid axis and assessed the ADCC in combination with IgA therapy. Our results showed that inhibiting both checkpoints led to greater tumor cell lysis than blocking either checkpoint alone, indicating that these checkpoint interactions act on different pathways (Figure 5A,B). We observed an additive enhancement of lysis in SK-BR-3 and A431 cells when both checkpoints were blocked, while for MDA-MB-468 cells, the enhancement was greater than an additive effect. Consistent with these findings, the efficacy of checkpoint inhibition appears to be more pronounced in tumor cells with lower tumor-associated antigen (TAA) expression levels, such as MDA-MB-468, compared to those with higher TAA expression levels, such as A431-HER2 and SK-BR-3 cells. These results demonstrate that both checkpoints can be blocked independently, but they can also be combined to further enhance IgA therapy.

### 3.5. Desialylation Improved IgA Therapy In Vivo

Next, we studied the impact of sialylation on tumor cells using a mouse model. Notably, mice express Siglec-E, which serves as the murine ortholog for Siglec-7 and -9 observed in humans [30]. We assessed the presence of Siglec-E ligands on our tumor cells (Appendix A) and found high levels of Siglec-E ligands on the human breast cancer cell line MDA-MB-468. Since wild-type mice do not express the Fc receptor for IgA, we used genetically modified mice expressing human FcαRI to investigate the role of sialylation on tumor cells in vivo. A short-term intraperitoneal (i.p.) xenograft model was established using MDA-MB-468 cells. However, considering the potential non-specific hydrolysis of neuraminidase on various cells in the peritoneum, we opted for an alternative approach to specifically desialylate the tumor cells. In vitro treatment of MDA-MB-468 cells with either DMSO (control) or sialyltransferase inhibitor (STinh) to disrupt sialic acid biosynthesis was performed three days prior to the inoculation. MDA-MB-468 cells were injected intraperitoneally three days later, along with target-negative Ba/F3 cells, followed by IgA or PBS treatment (Figure 6A). Desialylation of the tumor cells was confirmed through MAL II lectin binding analysis, which demonstrated reduced binding on the STinh-treated cells (Figure 6B). By comparing the ratios of the MDA-MB-468 and Ba/F3 cell counts, we determined the contribution of STinh treatment to the IgA therapy. Our results revealed that IgA anti-EGFR therapy led to a 50% reduction in tumor cells recovered from the peritoneum (Figure 6C and Appendix A). Furthermore, we observed that sialic acid removal with STinh significantly enhanced tumor cell killing, resulting in near-complete removal of tumor cells. Importantly, we confirmed that sialyltransferase treatment alone did not affect tumor killing. Ex vivo analysis of tumor cells using MAL II lectin confirmed the sustained low levels of surface sialic acids throughout the experiment (Figure 6D). These findings demonstrate that disrupting sialic acid biosynthesis in MDA-MB-468 cells significantly enhances the efficacy of IgA therapy.

## 4. Discussion

We investigated hypersialylation and CD47 expression in cancer cells and their role in modulating neutrophil cytotoxicity. Our findings confirm that CD47/SIRPα is an important myeloid checkpoint that regulates neutrophil killing of tumor cells. Furthermore, we investigated the role of additional inhibitory receptors expressed on neutrophils that inhibit neutrophil cytotoxicity outside the CD47/SIRPα axis.

Overexpression of CD47 has been strongly correlated with poor therapeutic outcomes in both solid and hematological malignancies [10,31,32]. We confirmed the expression of CD47 in a wide range of cancer cells, consistent with previous reports. Inhibition of CD47 has demonstrated potential to enhance neutrophil-mediated killing through IgA. However, despite CD47 blockade, certain cancer cells showed resistance to this therapeutic approach. These observations suggest that tumor cells possess additional myeloid checkpoint molecules that can suppress neutrophils and reduce their ADCC capacity. This is consistent with other reports of novel myeloid checkpoints. However, these studies generally focus on monocytes and macrophages rather than neutrophils [33,34,35]. In this study, we have shown how sialoglycans expressed on tumor cells affect ADCC by neutrophils induced by IgA antibodies.

Aberrant sialylation in tumor cells has been shown to facilitate immune evasion [36]. Tumor cells frequently express elevated levels of sialylated *O*-glycans, *N*-glycans, and gangliosides. Hypersialylation in cancer cells is often a consequence of altered sialyltransferase expression, among other factors. Sialyltransferase enzymes are crucial enzymes that add Neu5Ac to either galactose, *N*-acetylgalactosamine, or sialic acid [37]. The upregulation of these sialyltransferases is associated with tumor development characteristics such as hypoxia, DNA methylation, and Ras and c-Myc signaling [16]. 

Here, we found high levels of α2,3-linked sialic acids in our tumor cell panel. In SK-BR-3 cells, increased sialylation by uptake of Neu5Ac from the medium impaired IgA-mediated ADCC by neutrophils while removing these sialic acids by neuraminidases leads to improved ADCC. Desialylation by enzymatic hydrolysis of sialic acid or genetic knock out of GNE improved ADCC in the presence of an IgA antibody but did not affect the tumor in the absence of IgA, indicating the need for an activating cue to recruit neutrophils. Furthermore, desialylation of MDA-MB-468 cells improved IgA therapy in vivo and resulted in nearly complete tumor cell removal.

We examined the role of Siglec receptors in the immune response mediated by IgA antibodies. Our findings showed that neutrophils express Siglec-3, -5, -7, and -9, all of which signal through an ITIM and/or ITIM-like domain and activate downstream signaling pathways inhibiting immune cell function [18]. Through the use of recombinant Siglec-Fc proteins, we confirmed that our panel of cancer cell lines expressed ligands for Siglec-7 and -9.

Despite the low expression of Siglec-7 on neutrophils, we observed that when combined with high levels of Siglec-7 ligands on CFPAC-1 cells, it led to a suppressed neutrophil immune response. However, desialylation alone was insufficient to overcome neutrophil suppression. A possible explanation could be the concurrent high expression of CD47, which contributed to the dominant inhibitory signals. Importantly, these inhibitory signals, from both the sialoglycan/Siglec axis and the CD47/SIRPα axis, do not directly impact the mechanism of IgA-mediated ADCC. In collaboration with the Valerius group, we also observed improved neutrophil-mediated killing upon desialylation in combination with IgG therapeutics, suggesting a mechanism independent of the antibody isotype [23]. Instead, it is likely the balance between accumulating ITIM signals and ITAM signals that ultimately determines the level of immune activity. Moreover, the CD47/SIRPα axis is likely the more dominant inhibitory interaction. The presence of two ITIM domains on SIRPα, versus one ITIM domain in Siglec-7 and Siglec-9, could explain the difference [8]. As a result, SIRPα may induce a stronger inhibitory signal. By simultaneously targeting multiple immune checkpoints, we were able to overcome the resistance and rebalance the TME in favor of tumor lysis. Overall, our findings highlight the complexities of immune regulation through immune checkpoint molecules and the importance of considering various inhibitory axes.

Additionally, Siglec-9 is much higher expressed on neutrophils. Moreover, the majority of tumor cells in our panel expressed higher levels of ligands for Siglec-9 compared to Siglec-7. Using a blocking antibody against Siglec-9, we increased IgA ADCC capacity up to 80% of the efficacy of neuraminidase treatment in SK-BR-3 cells. This suggests that Siglec-9 is the more prominent inhibitory Siglec receptor, although Siglec-7 should not be disregarded. Siglec-9 expression is not limited to neutrophils; it is also found on macrophages, monocytes, and, to a lesser extent, B cells, NK cells, and a subset of T cells [38]. In macrophages, Siglec-9 ligation decreased pro-inflammatory cytokine production and increased IL-10 secretion, resulting in an M2 phenotype [26,39]. Similarly, Siglec-9 ligation decreased the ADCC capacity of NK cells [40,41]. The function of Siglec-9 in neutrophils during cancer therapy has received little attention up to this point. In inflammatory diseases such as acute septic shock or rheumatoid arthritis, Siglec-9 ligation was shown to induce apoptosis of neutrophils [42]. Furthermore, engagement of Siglec E, the mouse ortholog for Siglec-7 and -9, reduced inflammatory responses [43,44]. 

We observed high expression levels of α2,3-linked sialic acids in a panel of tumor cells, the preferred sialic acid for Siglec-9 [38]. Surprisingly, cancer cells that had more enriched α2,3-linked sialic acids did not correlate with increased Siglec-9-Fc binding. All Siglecs have a low affinity for galactose-linked α2,3- and α2,6-linked sialic acids. The binding of distinct Siglecs is heavily reliant not only on terminal sialic acids, but also on the subterminal glycan moiety, linkage orientation, and/or hydroxyl modification. Increased Siglec-9 ligands were found to be regulated, among other sialyltransferases, by ST3GAL1, which acts on core-1 *O*-glycans [26,45]. To date, MUC1 and MUC16 have been reported to be specific tumor antigens for Siglec-9 [22,26,46,47]. Siglec-9-Fc protein binding, rather than sialic acid detection with MAL II lectin, should provide a more accurate indication of Siglec-9-induced immune suppression.

We demonstrated that enzymatic desialylation successfully prevented the interaction between Siglec-7 and -9 and sialoglycans. In the clinic, however, such a non-specific approach is inapplicable. Therefore, we evaluated a Siglec-9-blocking antibody that gave comparable outcomes to neuraminidase treatment. Moreover, other methods could involve bispecific strategies to desialylate only tumor antigen-specific targets to prevent on-target adverse reactions [41,48]. However, additional research is required to determine the therapeutic efficacy of this antibody format. Furthermore, CD47 expression appeared to be more consistent across different cancer cell lines than Siglec-9 Fc binding. Certain sialyltransferases, such as the previously mentioned ST3GAL1, are frequently overexpressed in breast cancers and certain adenocarcinomas [49,50,51,52]. Therefore, targeting Siglec-9 may only be beneficial in certain tumor types, whereas targeting CD47 can have broad applicability across a wide range of both solid and hematological malignancies. However, as discussed earlier, considering a combination strategy that incorporates both Siglec-9 and CD47 blockade may potentially overcome resistance and enhance therapeutic outcomes in specific cases.

Checkpoint inhibition strategies hold promise as feasible options for resistant and advanced tumors. Our study demonstrated that cancer patient-derived PMNs remained susceptible to checkpoint inhibition treatments. Advanced tumors often overexpress checkpoint molecules as an immune evasion mechanism, emphasizing the importance of targeting these pathways.

## 5. Conclusions

In summary, we observed elevated levels of α2,3-linked sialic acids on tumor cells, which promote tumor progression by activating Siglec-7 and -9 on neutrophils. This study supports the notion that Siglecs are a novel class of neutrophil checkpoints. We demonstrated that the Siglec-9/Sialic acid axis can be effectively blocked by a Siglec-9 blocking antibody, and that the combination of this antibody with IgA therapy enhanced killing of tumor cells. Furthermore, this strategy complements CD47/SIRPα axis disruption, where blocking both checkpoints simultaneously improved the neutrophil killing capacity mediated by IgA antibodies. In conclusion, this study reveals the presence of at least two myeloid checkpoints and supports that inhibiting myeloid checkpoints is an effective strategy for overcoming therapeutic resistance to IgA therapy.

## Figures and Tables

**Figure 1 cancers-15-03405-f001:**
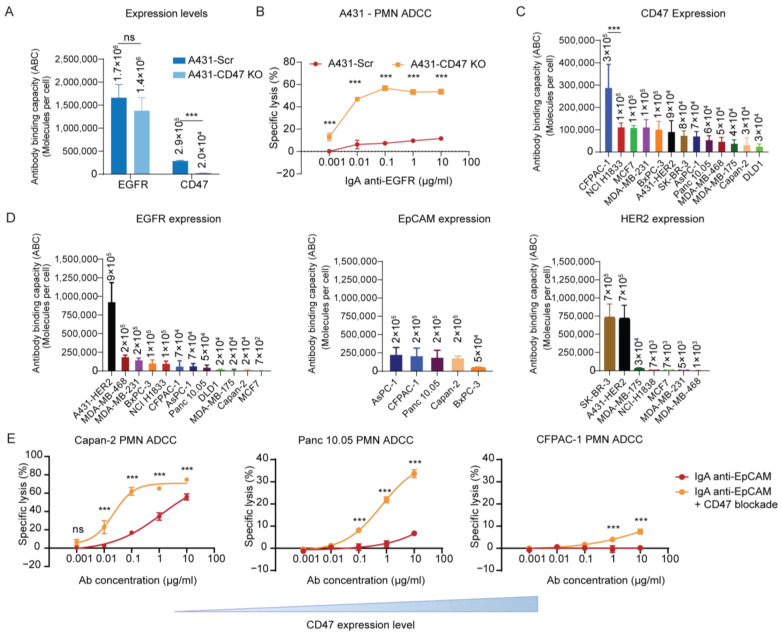
ADCC by IgA antibodies in combination with CD47 blockade. (**A**) CD47 and EGFR expression level on A431-scrambled (Scr, control) and A431-CD47 KO was determined by flow cytometry using 10 µg/mL target antibody with QIFIKIT analysis. Mean ± SD of antibody binding capacity (ABC) is shown of *n* = 3 independent experiments. ns > 0.05, *** *p* < 0.001 by Student’s *t* test (**B**) PMN-mediated ADCC against A431-Scr and A431-CD47KO by an IgA EGFR (cetuximab) antibody in concentrations ranging from 0 µg/mL to 10 µg/mL. Specific lysis was determined after 4 h in a ^51^Cr release assay. PMNs were co-cultured with the tumor cells at an E:T ratio of 40:1. The mean ± SD specific lysis of a technical triplicate of single donor is shown. At least *n* = 3 independent experiments are represented by 1 representative graph. (**C**,**D**) Absolute expression level of CD47 (C) and EGFR, EpCAM, HER2 (**D**) on a panel of tumor cell lines was determined by flow cytometry using 10 µg/mL target antibody with QIFIKIT. Mean ± SD of antibody binding capacity (ABC) is shown of *n* = 3 independent experiments. *** *p* < 0.001 as tested by one-way ANOVA followed by Tukey’s post-hoc test. (**E**) PMN-mediated ADCC against Capan-2, Panc 10.05, and CFPAC-1 by an IgA EpCAM (heING) antibody in concentrations ranging from 0 µg/mL to 10 µg/mL. CD47 was pre-blocked using SIRPα fusion protein at 10 µg/mL for at least 30 min. at RT at an E:T ratio of 40:1. The mean ± SD specific lysis of a technical triplicate of a single donor is shown. At least *n* = 3 independent experiments are represented by 1 representative graph. ns > 0.05, *** *p* < 0.001, by two-way ANOVA followed by Šidák post-hoc test.

**Figure 2 cancers-15-03405-f002:**
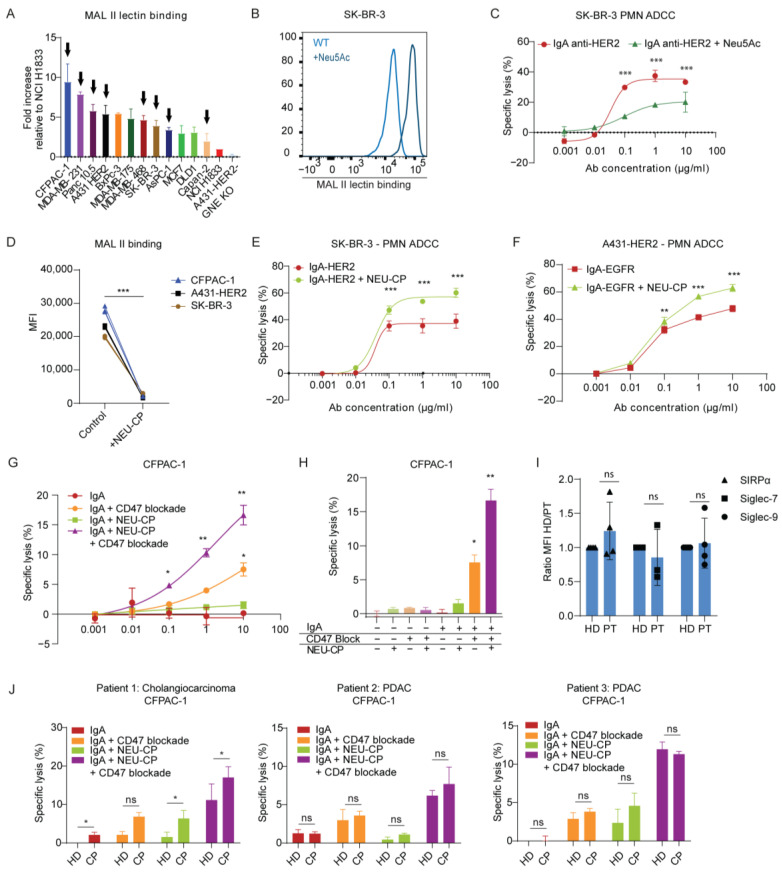
Hypersialylation on tumor cells correlated with decreased ADCC by neutrophils. (**A**) Expression of α2,3-linked sialic acids on a panel of tumor cells was determined using MAL II lectin binding by flow cytometry. Data were shown in fold change relative to geometric mean fluorescent intensity (MFI) of NCI H1833. Mean ± SEM is shown of *n* = 3 independent experiments. Cell lines of interest (high TAA expression levels) are indicated with an arrow. (**B**) Expression of α2,3-linked sialic acids on SK-BR-3 cells determined using MAL II are shown in a representative histogram. SK-BR-3 cells were treated for 48 h with 10 mM of Neu5Ac or PBS control (WT) in a humidified incubator at 37 °C, and 5% CO_2_. (**C**) PMN-mediated ADCC against SK-BR-3 cells by IgA HER2 (trastuzumab) in concentrations ranging from 0 µg/mL to 10 µg/mL. SK-BR-3 cells were treated for 48 h with 10 mM of Neu5Ac or PBS control in a humidified incubator at 37 °C, and 5% CO_2_. (**D**) Cells were treated with 0.1 U/mL neuraminidase for 1 h in a humidified shaking incubator at 125 rpm, 37 °C, and 8% CO_2_. Expression of α2,3-linked sialic acids was determined using MAL II lectin binding by flow cytometry and shown in MFI. *** *p* < 0.001 by Student’s *t*-test (**E**,**F**) Specific lysis of the NEU pre-treated SK-BR-3 and A431-HER2 cells was determined after 4 h in a ^51^Cr release assay. IgA HER2 (trastuzumab) was added in titrated concentrations. (**G**,**H**) PMN-mediated ADCC against CFPAC-1 by IgA EpCAM in titrated concentrations. Cells were pre-treated with NEU, and/or CD47 was pre-blocked using SIRPα fusion protein at 10 µg/mL for at least 30 min. at RT. (**H**) Lysis in the presence or absence of 10 μg/mL IgA EpCAM compared to experimental controls. (**I**) The expression levels of Siglec-7, Siglec-9, and SIRPα were compared between cancer patients and healthy donor controls. The data were presented as the ratio between the healthy donor control and the cancer patient. (**J**) PMNs obtained from a cancer patient were compared with PMNs isolated from a healthy donor in an ADCC assay. The ADCC capacity was evaluated against CFPAC-1 cells using IgA EpCAM at a concentration of 10 μg/mL. Tumor cells were subjected to pre-treatment with NEU, and/or CD47 was pre-blocked using SIRPα fusion protein at a concentration of 10 µg/mL for 30 min at RT. In total, three patients were included: patient 1 with cholangiocarcinoma, patient 2 with PDAC, and patient 3 with PDAC. HD = healthy donor, CP = cancer patient. In all ^51^Cr release ADCC assays unless stated otherwise, specific lysis was determined after 4 h. PMNs were co-cultured with the tumor cells at an E:T ratio of 40:1. The mean ± SD specific lysis of a technical triplicate of a single donor is shown. At least *n* = 3 independent experiments are represented by 1 representative graph. ns > 0.05, * *p* < 0.05 ** *p* < 0.01 *** *p* < 0.001, by two-way ANOVA followed by followed by Tukey’s post-hoc test.

**Figure 3 cancers-15-03405-f003:**
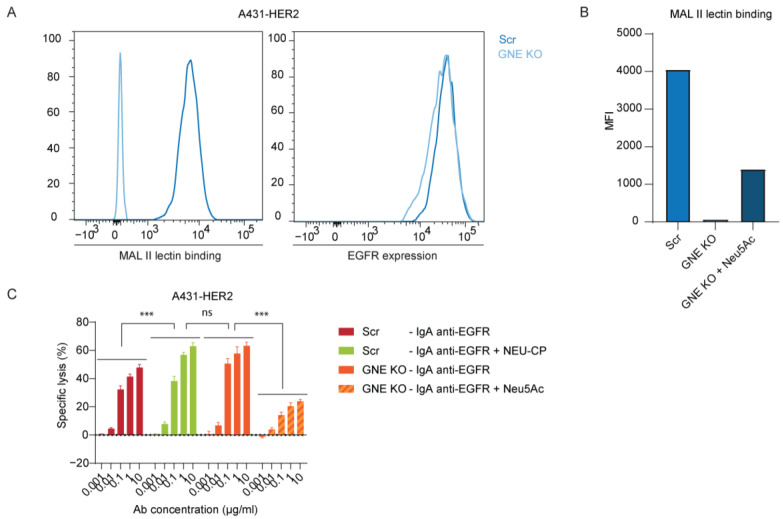
Knocking out GNE reduced sialylation of A431-HER2 and improved ADCC. (**A**) Expression of α2,3-linked sialic acids and EGFR on A431-HER2-Scr and A431-HER2-GNE KO cells by flow cytometry shown in a representative histogram. (**B**) Expression of α2,3-linked sialic acids on A431-HER2-Scr and A431-HER2-GNE KO cells determined using MAL II lectin by flow cytometry. A431-HER2-GNE KO cells were treated for 48 h with 10 mM of Neu5Ac in a humidified incubator at 37 °C and 5% CO_2_. Geometric MFI is presented. The figure displays a representative experiment out of three independent experiments (*n* = 3). (**C**) PMN-mediated ADCC against A431-HER2-Scr and A431-HER2-GNE KO cells by IgA EGFR (cetuximab) in concentrations ranging from 0 µg/mL to 10 µg/mL. Cells were pre-treated with 0.1 U/mL neuraminidase for 1 h in a humidified shaking incubator at 125 rpm, 37 °C, and 8% CO_2_ or with 10 mM of Neu5Ac for 48 h in a humidified incubator at 37 °C and 5% CO_2_. Specific lysis was determined after 4 h in a ^51^Cr release assay. PMNs were co-cultured with the tumor cells at an E:T ratio of 40:1. The mean ± SD specific lysis of a technical triplicate of a single donor is shown. At least *n* = 3 independent experiments are represented by 1 representative graph. ns > 0.05, *** *p* < 0.001, by two-way ANOVA followed by Tukey’s post-hoc test.

**Figure 4 cancers-15-03405-f004:**
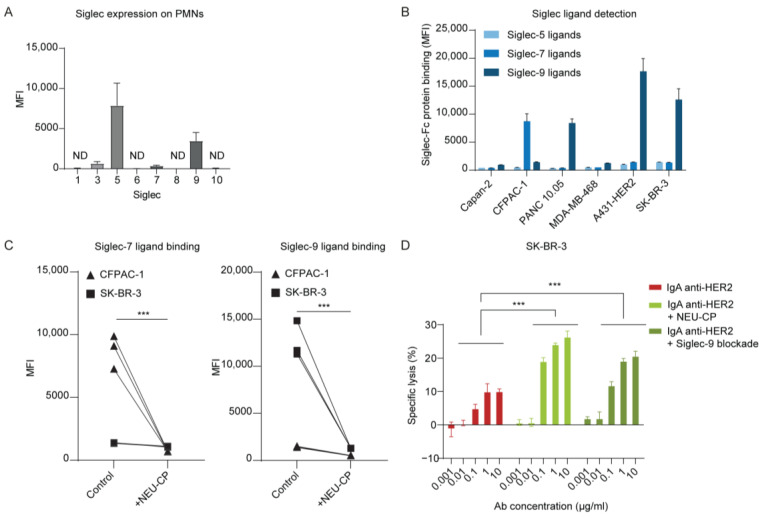
Reduction in Siglec ligands improved ADCC by IgA antibodies. (**A**) Siglec expression level on PMNs was determined by flow cytometry using conjugated Siglec-targeting antibodies. Mean ± SD of geometric MFI is shown for *n* = 3 independent experiments with 3 different healthy donors. ND = not detected (**B**) Detection of Siglec-5, -7, and -9 ligands on a panel of tumor cells with 10 μg/mL Siglec-Fc proteins by flow cytometry. Mean ± SEM of geometric MFI is shown of *n* = 3 independent experiments. (**C**) Binding of CFPAC-1 and SK-BR-3 cells to Siglec-7 and Siglec-9 Fc proteins before and after neuraminidase treatment assessed with flow cytometry. *** *p* < 0.001 by Student’s *t*-test (**D**) PMN-mediated ADCC against SK-BR-3 by IgA HER2 (trastuzumab) in titrated concentrations. Siglec-9 was pre-blocked using Siglec-9 blocking antibody at 20 µg/mL for at least 30 min at RT. Specific lysis was determined after 4 h in a ^51^Cr release assay. PMNs were co-cultured with the tumor cells at an E:T ratio of 40:1. The mean ± SD specific lysis of a technical triplicate of a single donor is shown. At least *n* = 3 independent experiments are represented by 1 representative graph. *** *p* < 0.001, by two-way ANOVA followed by Tukey’s post-hoc test.

**Figure 5 cancers-15-03405-f005:**
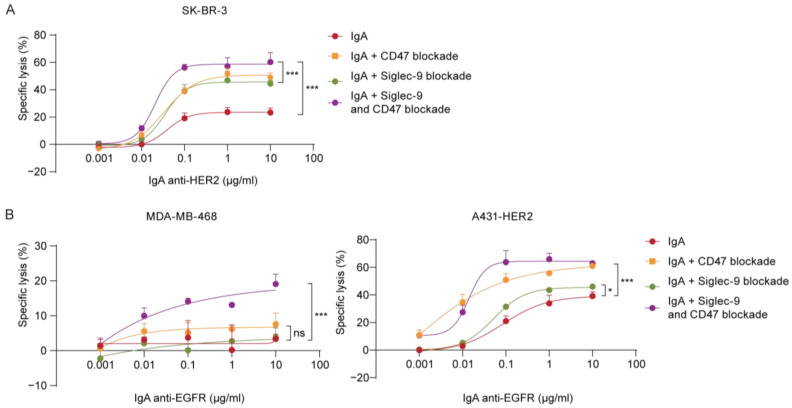
Combining checkpoint inhibitors targeting CD47 and Siglec-9 enhanced IgA-mediated killing. PMN-mediated ADCC was assessed in a 4 h ^51^Cr release assay using IgA concentrations ranging from 0 µg/mL to 10 µg/mL. PMNs were co-cultured with the tumor cells at an E:T ratio of 40:1. Siglec-9 was pre-blocked using Siglec-9-blocking antibody at 20 µg/mL, and/or CD47 was pre-blocked using SIRPα fusion protein at 10 µg/mL. Both antibodies were incubated for at least 30 min at RT. Specific lysis of (**A**) SK-BR-3 (HER2 high) cells and (**B**) MDA-MB-468 (EGFR-low) and A431-HER2 (EGFR-high) was determined and shown as mean ± SD of a technical triplicate of a single donor. At least *n* = 3 independent experiments are represented by 1 representative graph. ns > 0.05, * *p* < 0.01, *** *p* < 0.001, by two-way ANOVA followed by a Dunnett post-hoc test.

**Figure 6 cancers-15-03405-f006:**
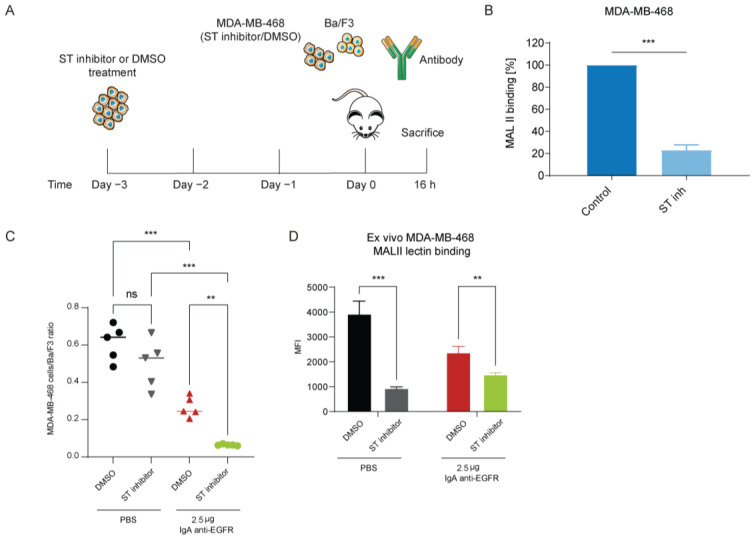
Desialylation of MDA-MB-468 cells improved IgA therapy in vivo. (**A**) Schematic time overview of the in vivo short i.p. xenograft model. MDA-MB-468 cells were treated in vitro with a sialyltransferase inhibitor or DMSO on day -3. Tumor cells were inoculated 3 days after treatment on day 0, and IgA therapy was administered on the same day. Mice were sacrificed 16 h after injection, and a peritoneal lavage was performed. (**B**) MAL II binding on PBS or sialyltransferase inhibitor-treated MDA-MB-468. Data are shown as percentage of MAL II binding. *** *p* < 0.001 by Student’s *t*-test (**C**) The ratio of MDA-MB-468 and Ba/F3 in the peritoneal lavage was determined by flow cytometry. Mean ± SEM is shown of *n* = 5 mice. ns > 0.05, ** *p* < 0.01, *** *p* < 0.001, by one-way ANOVA followed by Bonferroni post-hoc test. (**D**) α2.3-linked sialic acid expression level was determined on MDA-MB-468 cells recovered from the peritoneum. Mean ± SD of geometric MFI is shown for *n* = 5 mice. ** *p* < 0.01, *** *p* < 0.001, by two-way ANOVA followed by Šidák post-hoc test.

## Data Availability

The data generated in this study are available upon request from the corresponding author.

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
