# Peer review of "Sialic Acids on Tumor Cells Modulate IgA Therapy by Neutrophils via Inhibitory Receptors Siglec-7 and Siglec-9"

_cancers, 2023, doi:10.3390/cancers15133405_

Round 1

Reviewer 1 Report

1.    The study of Chilam Chan and colleagues investigates the effect of checkpoint inhibition on IgA therapy beyond the CD47/Signal regulatory protein alpha ( SIRPα) axis focusing on neutrophils. The CD47/SIRPα axis is a known key regulator in tumor growth and its inhibition in combination with IgA therapy has shown promise, but no complete tumor eradication was  observed, indicating the presence of other checkpoints. One of these mechanisms could be found in aberrant sialylation of cancer cells, which function as negative regulator of immune cells . Authors investigated  hypersialylation on the tumor cell surface as a potential myeloid checkpoint, and found that hypersialylated cancer cells inhibit neutrophil-mediated tumor killing through interactions with sialic acid-binding immunoglobulin-like lectins (Siglecs). To enhance antibody-dependent cellular cytotoxicity (ADCC) using IgA as therapeutic, they explored strategies to disrupt the interaction between tumor cell sialoglycans and Siglecs expressed on neutrophils and identified Siglec-9 as the primary inhibitory receptor, with Siglec-7 also playing a role to a lesser extent. They demonstrated that blocking Siglec-9 enhanced IgA-mediated ADCC by neutrophils.

Authors conclude that their findings suggest that  Siglecs are a novel class of neutrophil checkpoints. The blockade of Siglec-9/Sialic acid axis combined with IgA therapy enhanced killing of tumor cells. A combination of CD47 blockade and desialylation may be necessary to optimize cancer immunotherapy. Blocking both checkpoints simultaneously improved the neutrophil killing capacity mediated by IgA antibodies. The results support the notion that inhibiting myeloid checkpoints is an effective strategy for overcoming therapeutic resistance to IgA therapy.

 This study is interesting and highlights the importance of multiple myeloid checkpoints in the resistance of tumor cells to immune intervention. However, future studies will be necessary to determine its potential clinical applicability and the plausibility of clinical trials

 Some issues should be addressed:

1.      Figure 2 I. PMN from patient with colangiocarcinoma display more ADCC activity than those from healthy donor (*p < 0.05). Authors should comment this issue.  What about ADCC activity of PMN from the other cancer patient with PDAC? . Moreover only two cancer patients to evaluate PMN activity is too small a number

2.      Figure 5 C. For A431-HER2  target cells, there is no increase of lysis induced by IgA blocking CD47 in comparison to combining  checkpoint inhibitors targeting CD47 and Siglec-9. Please, comment.

3.      References section: number of pages is missed in ref. 2, 4. Ref 33:  the number of volume is wrong. The name of journal of ref. 37, 39, 46 should be written abbreviated

Author Response

Thank you for the insightful comments. In response to the raised concerns, we would like to address the following points:

Some issues should be addressed:

Figure 2 I. PMN from patient with colangiocarcinoma display more ADCC activity than those from healthy donor (*p < 0.05). Authors should comment this issue.  What about ADCC activity of PMN from the other cancer patient with PDAC? . Moreover only two cancer patients to evaluate PMN activity is too small a number

In response to this concern, we have made efforts to increase the patient inclusion size. We were able to measure the ADCC activity from three patients. Taken together, our analysis suggests no differences between the ADCC activity observed with PMNs from healthy donors and those obtained from cancer patients. These findings suggest that the ADCC capacity of PMNs were not altered by the disease and PMNs remained susceptible to checkpoint inhibition treatments and IgA therapy.

Figure 5 C. For A431-HER2  target cells, there is no increase of lysis induced by IgA blocking CD47 in comparison to combining  checkpoint inhibitors targeting CD47 and Siglec-9. Please, comment.

This is a valid point that has also been highlighted by one of the other reviewers. Figure 5 demonstrates that at higher antibody concentrations, there is reduced involvement of the checkpoint molecules. Our hypothesis proposes that IgA drives ITAM signaling, while the interactions with the investigated checkpoint molecules induce ITIM signaling. Consequently, as the concentration of IgA increases (assuming a sufficiently high expression level of the tumor antigen), the dominance of ITAM signaling becomes more prominent, leading to a decrease in the significance of the checkpoint interactions. Consistent with this hypothesis, we observed a greater significance of the checkpoint inhibitors in Figure 5B, left panel, as the antigen expression level for MDA-MB-468 is much lower, and thus the ITAM signaling is less pronounced.

References section: number of pages is missed in ref. 2, 4. Ref 33:  the number of volume is wrong. The name of journal of ref. 37, 39, 46 should be written abbreviated
Thank you for pointing this out, this has now been corrected in the revised manuscript.

Reviewer 2 Report

Leusen's group has made significant contributions to the field of CD47/SIRPα-axis research. This paper investigates the role of hypersialylation as a potential myeloid checkpoint in tumor cells and its effect on IgA therapy by neutrophils. The authors discovered that hyper-sialylated cancer cells inhibit neutrophil-mediated tumor lysis through interactions with Siglecs, with Siglec-9 identified as the primary inhibitory receptor. They also explored strategies to disrupt the interaction between tumor cell sialoglycans and Siglecs expressed on neutrophils, with the aim of enhancing ADCC using IgA as a therapeutic agent. This paper provides valuable insights into the potential of targeting Siglecs in IgA therapy for cancer and emphasizes the need for a multi-checkpoint blocking approach.

Comments:

The paper offers a detailed exploration of the role of hypersialylation as a potential myeloid checkpoint in tumor cells and its impact on IgA therapy by neutrophils. This novel contribution to the field showcases an innovative approach by examining the potential of Siglec targeting in IgA therapy for cancer and emphasizes the significance of a multi-checkpoint blocking strategy. Furthermore, the paper is well-structured and includes a comprehensive review of relevant literature, making it highly valuable to readers.

I believe this paper has the potential to be published in cancers after undergoing several revisions.

Major points:

(1)   The paper lacks specific details regarding the implementation of the study, making it challenging to reproduce. I strongly recommend that the authors provide a summary cartoon illustrating the main interactions between tumor cells, neutrophils, and IgA therapy.

(2)   While the study sheds light on the potential of targeting Siglecs in IgA therapy, its impact is limited by the absence of a comparison with other well-established baselines in the field. Is the PD-L1 and CTLA4 pathway involved in this process? Does CD22, another "Don't eat me" signaling protein, play a role in this interaction? It is important to include experiments or discuss these factors in the Discussion section.

Minor points:

1、 Please verify the meaning of the "†" symbol in the author list.

2、 In line 118-119, it is stated, "Sialic acids on target cells were hydrolyzed by pre-treatment with 0.1 U/ml of neuraminidase (NEU) for 1 hour in a shaking incubator at 125 rpm and 37 ºC." Please provide additional details about the reaction buffer used.

3、 In line 145, the lectin "Maackia Amurensis Lectin II" should be italicized for the species name "Maackia Amurensis."

4、 In line 161, "100.000 cells were stained with antibody ..." Please verify the punctuation (comma) in this sentence.

5、 In line 196, "ul" should be written as "μl."

6、 In line 196, "9 M" cells, please clarify the meaning of "M."

7、 In Figure 1D, please provide an explanation for the "μ" symbol in the y-axis annotation for "EpCAM expression."

8、 In line 269, "ug" should be written as "μg."

9、 In line 277, there should be a space in the word "sialylatedsuggesting."

10、         In Figure 2I, four colors are not annotated in the color key.

Author Response

Thank you for critically reading our manuscript and helping us to improve it.We would like to address the following concerns:

Major points:

(1)   The paper lacks specific details regarding the implementation of the study, making it challenging to reproduce. I strongly recommend that the authors provide a summary cartoon illustrating the main interactions between tumor cells, neutrophils, and IgA therapy.

To improve the understanding of our study and facilitate reproducibility, we have added a graphical abstract to illustrate the main interactions between tumor cells, neutrophils and IgA therapy.

(2)   While the study sheds light on the potential of targeting Siglecs in IgA therapy, its impact is limited by the absence of a comparison with other well-established baselines in the field. Is the PD-L1 and CTLA4 pathway involved in this process? Does CD22, another "Don't eat me" signaling protein, play a role in this interaction? It is important to include experiments or discuss these factors in the Discussion section.

We appreciate your consideration and understand the importance of comparing our findings with other well-established checkpoints in the field. However, we would like to clarify that our study specifically focuses on the role of myeloid cells, particularly neutrophils, in the context of Siglec targeting. While PD-L1, CTLA-4, and CD22 are indeed significant checkpoints, they are primarily involved in T cell and B cell function. Their direct relevance to our research question is beyond the scope of this manuscript. Instead, we have chosen to compare Siglec-7 and -9 with another well-established myeloid checkpoint, the CD47/SIRPa axis. This allows us to focus on checkpoint interactions modulating neutrophil responses.

Minor points:

1 Please verify the meaning of the "†" symbol in the author list.
† indicates authors that have contributed equally, this has now been added to the text.

2 In line 118-119, it is stated, "Sialic acids on target cells were hydrolyzed by pre-treatment with 0.1 U/ml of neuraminidase (NEU) for 1 hour in a shaking incubator at 125 rpm and 37 ºC." Please provide additional details about the reaction buffer used.
Serum-free RPMI medium was used during the incubation. This information is now added to the revised manuscript.

3 In line 145, the lectin "Maackia Amurensis Lectin II" should be italicized for the species name "Maackia Amurensis."
This is corrected in the revised manuscript.

4 In line 161, "100.000 cells were stained with antibody ..." Please verify the punctuation (comma) in this sentence.
This is corrected in the revised manuscript.

5 In line 196, "ul" should be written as "μl."
This is corrected in the revised manuscript.

6 In line 196, "9 M" cells, please clarify the meaning of "M."
This is changed to “9 million” in the revised manuscript.

7 In Figure 1D, please provide an explanation for the "μ" symbol in the y-axis annotation for "EpCAM expression."
This is an error, and is now removed.

8 In line 269, "ug" should be written as "μg."

      We addressed this in the revised manuscript.

9 In line 277, there should be a space in the word "sialylatedsuggesting."

This is corrected in the revised manuscript.

10         In Figure 2I, four colors are not annotated in the color key.
Colors are changed, and the difference between healthy donor and cancer patient is annotated on the x-axis.

Reviewer 3 Report

In this manuscript Chan and colleagues report that cancer cell sialylation patterns inhibit neutrophil phagocytic function. They identify Siglec-9 and Siglec-7 as sialoglycan partners on neutrophils. They demonstrate that blocking Siglec-9 or desialylation enhances IgA-mediated ADCC by neutrophils.

Overall, this is a well-designed study and the mansurcipt is well-written. However, this work does not seem novel. Indeed, the same group in collaboration with Dr Valarius published a very similar study, with overlapping data and messages, this month: https://www.frontiersin.org/articles/10.3389/fimmu.2023.1178817/full

This raises major concerns and undermines the novelty of the data presented. This paper is not cited in this manuscript.

Additionally, I would have liked to see the combinational therapy with IgA and CD47/Siglec-9 mAbs to be confirmed in their in vivo system.

Please see minor comments below:

1-      Line 157: Thomas Valarius is a co-author and should not be named here.

2-      Line 259: delete ‘The’ in the beginning of sentence.

3-      Figure 5: In 2/3 of cell lines tested, the effects of combination therapy appear significant but are marginal. One would have expected a more pronounced additive effect with CD47 and Siglec-9 blockade. Can authors comment on this?

4-      The in vitro data are with human neutrophils. With respect to Figure 6, which murine Siglecs bind to sialoglycans on human cancer cell?

Author Response

In this manuscript Chan and colleagues report that cancer cell sialylation patterns inhibit neutrophil phagocytic function. They identify Siglec-9 and Siglec-7 as sialoglycan partners on neutrophils. They demonstrate that blocking Siglec-9 or desialylation enhances IgA-mediated ADCC by neutrophils.

Overall, this is a well-designed study and the mansurcipt is well-written. However, this work does not seem novel. Indeed, the same group in collaboration with Dr Valarius published a very similar study, with overlapping data and messages, this month: https://www.frontiersin.org/articles/10.3389/fimmu.2023.1178817/full

This raises major concerns and undermines the novelty of the data presented. This paper is not cited in this manuscript.

Additionally, I would have liked to see the combinational therapy with IgA and CD47/Siglec-9 mAbs to be confirmed in their in vivo system.

We greatly appreciate your valuable feedback. Indeed, in collaboration with Dr. Valerius we published on a similar topic, where we focused on neutrophil cytotoxicity induced by IgG antibodies, to be more precise, IgG1 and IgG2 antibodies. This manuscript has been accepted for publication during the review process of the present manuscript, so it can now be added to the references, thank you for bringing this up.

The present manuscript interrogates the neutrophil function induced by IgA and the combination with other well-established myeloid checkpoint, CD47/SIRPa. We clarified this in the revised manuscript and the work with Dr. Valerius is now cited.

The suggestion to confirm the combinational therapy with IgA and CD47/Siglec-9 mAbs is interesting. However, it is important to note that mice do not express the human Siglec-9 receptor but instead express a mouse ortholog, Siglec E. As a result, it would not have been feasible to directly examine this specific targeting approach with Siglec-9 blocking antibodies in our current in vivo systems. Although this experiment was not possible within the scope of our manuscript, it would be very interesting to explore the proposed approach in a Siglec-7/9 humanized mouse model in future projects.  

Please see minor comments below:

1-      Line 157: Thomas Valarius is a co-author and should not be named here.
Indeed that is the case so we corrected this in the revised version of the manuscript.

2-      Line 259: delete ‘The’ in the beginning of sentence.
This was addressed in the revised manuscript.

3-      Figure 5: In 2/3 of cell lines tested, the effects of combination therapy appear significant but are marginal. One would have expected a more pronounced additive effect with CD47 and Siglec-9 blockade. Can authors comment on this?
This concern was also raised by another reviewer. Our findings in Figure 5, demonstrate that as antibody concentrations increase, the impact of checkpoint inhibitors is lowered. In line with our hypothesis, which suggests that IgA drives ITAM signaling while the checkpoint molecules investigated in this manuscript induce ITIM signaling, we propose that the dominance of ITAM signaling becomes more pronounced with higher IgA concentrations and higher tumor target antigen expression levels, resulting in less significant checkpoint interactions. Notably, our results in Figure 5B, left panel, support this hypothesis, as the lower antigen expression level in MDA-MB-468 cells reduced the prominence of ITAM signaling, thereby resulting in a greater significance of the checkpoint inhibitors in this cell line.

4-      The in vitro data are with human neutrophils. With respect to Figure 6, which murine Siglecs bind to sialoglycans on human cancer cell?
Based on previous studies mentioned in the literature, it has been suggested that Siglec E serves as the murine ortholog for Siglec 7 and 9. We have added this remark and reference in the revised manuscript. Additionally, we have included a supplementary figure in our manuscript that depicts the levels of Siglec E ligands present on the human cancer cells in our panel.

Round 2

Reviewer 2 Report

Thank you for your prompt response and the revisions you have made based on the  comments. I am pleased to see that you have addressed all my concerns raised during the review process. The modifications you have implemented have significantly strengthened the clarity and impact of the manuscript.

I have no further question, and I am delighted to recommend the current version of your manuscript for publication in cancers journal.

Reviewer 3 Report

Thank you for revising the manuscript. I have no further comments.